# Regression-Based Normative Data for the Montreal Cognitive Assessment (MoCA) and Its Memory Index Score (MoCA-MIS) for Individuals Aged 18–91

**DOI:** 10.3390/jcm11144059

**Published:** 2022-07-13

**Authors:** Roy P. C. Kessels, Nathalie R. de Vent, Carolien J. W. H. Bruijnen, Michelle G. Jansen, Jos F. M. de Jonghe, Boukje A. G. Dijkstra, Joukje M. Oosterman

**Affiliations:** 1Donders Institute for Brain, Cognition and Behaviour, Radboud University, 6525 GD Nijmegen, The Netherlands; michelle.jansen@donders.ru.nl (M.G.J.); joukje.oosterman@donders.ru.nl (J.M.O.); 2Vincent van Gogh Institute for Psychiatry, Center of Excellence for Korsakoff and Alcohol-Related Cognitive Disorders, 5803 DN Venray, The Netherlands; carolienzeetsen@outlook.com; 3Klimmendaal Rehabilitation Specialists, 6813 GG Arnhem, The Netherlands; 4Tactus Addiction Care, 7400 AD Deventer, The Netherlands; 5Department of Medical Psychology and Radboudumc Alzheimer Center, Radboud University Medical Center, 6500 HB Nijmegen, The Netherlands; 6Department of Psychology, University of Amsterdam, 1018 WS Amsterdam, The Netherlands; n.r.devent@uva.nl; 7Nijmegen Institute for Scientist-Practitioners in Addiction (NISPA), Radboud University, 6525 GD Nijmegen, The Netherlands; boukje.dijkstra@outlook.com; 8ZONL, 8303 BX Emmeloord, The Netherlands; jdejonghe.expertise@gmail.com; 9Novadic-Kentron, Addiction Care Center, 5261 LX Vught, The Netherlands

**Keywords:** neuropsychological assessment, cognitive screening, aging, normative data, cognitive disorders

## Abstract

(1) Background: There is a need for a brief assessment of cognitive function, both in patient care and scientific research, for which the Montreal Cognitive Assessment (MoCA) is a psychometrically reliable and valid tool. However, fine-grained normative data allowing for adjustment for age, education, and/or sex are lacking, especially for its Memory Index Score (MIS). (2) Methods: A total of 820 healthy individuals aged 18–91 (366 men) completed the Dutch MoCA (version 7.1), of whom 182 also completed the cued recall and recognition memory subtests enabling calculation of the MIS. Regression-based normative data were computed for the MoCA Total Score and MIS, following the data-handling procedure of the Advanced Neuropsychological Diagnostics Infrastructure (ANDI). (3) Results: Age, education level, and sex were significant predictors of the MoCA Total Score (Conditional *R*^2^ = 0.4, Marginal *R*^2^ = 0.12, restricted maximum likelihood (REML) criterion at convergence: 3470.1) and MIS (Marginal *R*^2^ = 0.14, REML criterion at convergence: 682.8). Percentile distributions are presented that allow for age, education and sex adjustment for the MoCA Total Score and the MIS. (4) Conclusions: We present normative data covering the full adult life span that can be used for the screening for overall cognitive deficits and memory impairment, not only in older people with or people at risk of neurodegenerative disease, but also in younger individuals with acquired brain injury, neurological disease, or non-neurological medical conditions.

## 1. Introduction

The assessment of cognitive function in clinical practice is crucial as part of the diagnostic work-up of patients with acquired brain injury, neurodegenerative disease, and other medical conditions and has become increasingly important as an outcome measure in clinical trials and experimental research. Neuropsychological assessment is the gold standard for measuring cognitive performance in various domains, such as memory, orientation, executive function, language, and visuoconstructive ability [1]. However, neuropsychological assessments, in general, are time-consuming, costly, and require expertise in administration, scoring, and interpretation. As a result, tools that enable a shorter examination of cognitive performance have been extensively studied over the years. This has resulted in cognitive screening instruments that have gained widespread use yet have poor psychometric properties, such as the Mini-Mental State Examination [2]. Other cognitive screening tools have been developed with acceptable psychometric properties, such as the Addenbrooke’s Cognitive Examination (currently in its third edition, the ACE-III) or the Revised Cambridge Cognitive Assessment (CAMCOG-R). However, these screeners have been predominantly developed for the detection of dementia, lack the sensitivity to detect more subtle cognitive impairments, and may not be available in all territories [3]. The Montreal Cognitive Assessment (MoCA) has been developed to overcome these shortcomings, with the aim to better detect individuals with mild cognitive impairment (MCI) [4]. In short, the MoCA consists of several items addressing the domains of visuoconstruction, executive function, memory, language, orientation, and attention that can be easily administered and scored (with a maximum score of 30) and that also comes with parallel versions to minimize test-specific practice effects (www.mocatest.org (accessed on 20 June 2022)).

Since its introduction, the MoCA has gained international popularity, as its developers have been very successful in introducing culture-sensitive and language-specific international versions. Also, strict harmonization guidelines have been maintained to enable international comparison. Moreover, its psychometric properties have been studied extensively, showing a good test–retest reliability [5,6]. However, results on the validity of the instrument’s cut-off scores are somewhat mixed. Traditionally, the MoCA employs a cut-off score for the detection of individuals with dementia or MCI, as compared to cognitively unimpaired older adults. Originally, a cut-off score of <26 was established for detecting both MCI and Alzheimer’s dementia in healthy older adults, with a sensitivity of 90–100% and a specificity of 87% [4]. However, with regard to this cut-off score, several studies demonstrated a lower sensitivity and specificity for detecting MCI (e.g., 72%/73% [7]; 85%/81% [8]; 96%/58% [9]). A low specificity has also been reported for the detection of dementia [10]. One explanation for these mixed findings may lie in the demographic variables of the participants, notably educational attainment and age, which may vary considerably across studies and geographical regions. In the original MoCA study [4], no effect of age on the MoCA score was reported, whereas individuals with 12 years of education or less performed worse on the MoCA, resulting in an education adjustment of 1 additional point for those with 12 years of education or less. However, this original study only included a relatively small sample of older adults, limiting the participants’ age range. Later studies using samples with a wider age range showed that age, sex, and/or education level were significant predictors of the MoCA total score [6,11].

The need for a more fine-grained adjustment for age, sex, and education in interpreting an individual’s performance of the MoCA has become even more important as the MoCA is increasingly being used in a variety of patient samples. While originally developed for the detection of MCI and dementia due to Alzheimer’s disease, the MoCA has been studied to assess cognitive deficits in patients with a variety of medical conditions [12], including HIV [13], Parkinson’s disease [14], multiple sclerosis [15], stroke [16], frontotemporal dementia [17], substance-related cognitive disorders [18,19], cardiac arrest [20], fibromyalgia [21], Huntington’s disease [22], syncope and unexpected falls [23], cerebellar disease [24], schizophrenia [25], sickle cell disease [26], type 2 diabetes [27], and COVID-19 [28]. Many of these conditions affect younger adults, who may perform above established cut-off scores based on samples of older individuals even when cognitive impairment is present. Alternatively, individuals with low education levels (or even poor literacy) may perform below the cut-off score in the absence of cognitive deficits. This highlights the importance of age- and education-adjusted normative data for the MoCA total score in individuals below the age of 65. So far, several studies have established demographically-adjusted normative data for different language versions of the MoCA (see Table 1 for an overview). However, many studies that present normative to date focused on older adults exclusively, some studies have included only small samples, and no studies have examined samples from Northwest Europe.

In addition, the Memory Index Score (MoCA-MIS) is a relatively recent addition to the standard MoCA (with a maximum score of 15), which extends the classic free-recall memory subtest with a cued-recall and multiple-choice recognition test [61]. The MoCA-MIS has been studied in individuals with MCI, showing that a MoCA-MIS cut-off of <7, in addition to a total MoCA cut-off score of < 20, resulted in a 90.5% conversion rate for Alzheimer’s dementia after 18 months. Others [62] have demonstrated that the MoCA-MIS was able to distinguish cognitively unimpaired older adults from MCI patients to the same extent as a story recall test. Additionally, relative to other MoCA indices, the MoCA-MIS demonstrated the largest group difference between patients with alcohol Korsakoff’s syndrome, patients with alcohol-related cognitive disorder other than Korsakoff’s syndrome, and uncomplicated alcoholics [63]. Although the study by Kaur et al. [62] administered the MoCA-MIS in 2205 older adults (mean age = 72.7), they did not provide age and education-adjusted scores for this large sample (apart from a non-informative group mean of 12.2, SD = 2.8). Apolinario et al. [30] presented regression-based normative data for 597 cognitively unimpaired Brazilians aged 50 to 90. However, 81% (*n* = 484) of the participants in their sample completed only 10 or fewer years of formal education, making this sample not representative of the population of many other countries. Moreover, data on the MoCA-MIS for individuals younger than 50 are lacking altogether.

The present study sets out to compute age- and education-based normative data for the Dutch language version of the MoCA and the MoCA-MIS for individuals aged 18–91 using a demographically-adjusted regression-based normative approach, the validity of which has long been established for neuropsychological tests [64,65]. Such normative data facilitate the use of the MoCA as a short but valid cognitive screen in patient samples other than older adults with MCI or dementia.

## 2. Materials and Methods

### 2.1. Participants

A total of 825 healthy participants were included in this study (Table 2 shows the demographics of all participants).

All participants took part in research projects as healthy volunteers. Of these, 363 were volunteers in the ABRIM cohort who participated in aging research across the lifespan at the Donders Centre for Cognition, and 210 individuals took part as volunteers in a study on the psychometric properties of the MoCA [6]. In addition, healthy control samples from various other studies were included: 45 individuals participated in [68,69], 27 in [70], 60 people participated as controls in [71], 25 in a study by Sutter and colleagues [72], and 65 in studies by De Jonghe et al. (including [7,23]). Exclusion criteria included age younger than 18, the presence of any condition with a profound impact on the brain and cognitive health beyond normal aging, such as psychiatric disorders (e.g., major depressive disorder, bipolar disorder, schizophrenia), neurological conditions (e.g., dementia, history of stroke, epilepsy), substance use disorders, or a history of other major health conditions that could impact cognition (e.g., history of brain tumor). Care was taken to include individuals representative of all age groups and education levels, with a balanced sex distribution. Education level was scored in 7 categories, based on the Dutch educational system [66], which is similar to UNESCO’s ISCED scale [73], in which 1 reflects less than primary school and 7 an academic degree. These levels were then categorized as low (levels 1–4), average (level 5), or high (levels 6–7) for descriptive purposes only. All participants were fluent in Dutch.

### 2.2. Procedure

All participants were administered the authorized paper-and-pencil Dutch version of the Montreal Cognitive Assessment 7.1 (www.mocatest.org (accessed on 20 June 2022)). The optional cued-recall and multiple-choice memory test items were administered in *n* = 184, enabling the calculation of the MIS (maximum score = 15). Test administration and scoring were performed by trained psychologists in accordance with the test instructions and scoring manual. The MoCA Total Score reflects the education-uncorrected Total Score (maximum = 30). Higher scores reflect better performance.

### 2.3. Data Processing and Analyses

As the data are part of the larger Advanced Neuropsychological Diagnostics Infrastructure (ANDI), the standardized data-handling procedure for ANDI was followed. More details about this process can be found elsewhere [74], but we will provide a brief summary of all the data processing steps here.

First, all data from different sources were combined in a single data file. For each source, a unique identifier was added. After merging the data, we checked whether all recorded values were valid observations for healthy individuals. We defined ‘extreme borders’, with, on one side, the high border (set as the maximum possible score) and on the other, the low border, which reflected the lowest possible score that could be obtained from a person given they are indeed cognitively healthy. Scores exceeding the high border (coding errors) or low border (indicative of severe pathology) were removed from the dataset in order not to overestimate the variance. Age was coded as male = 0 and female = 1; education was coded in the aforementioned 7 categories.

Subsequently, we fitted a multi-level regression model to determine which scores were demographically-corrected outliers. Not all outliers can be found by looking at a single criterion value. Neuropsychological test scores depend, to some extent, on the demographic characteristics of the person. For this reason, we wanted to parse out the effects of age, sex, and level of education and determine which scores were abnormal given this demographic correction. Because the data from both the MoCA total score and MoCA MIS originated from different sources, a multi-level model was fit to consider the differences between studies [75]. Although level of education is an ordinal scale, we treated it as an interval scale and estimated the linear effect of education to avoid estimating separate parameters for all levels of education. To determine which demographic corrections were necessary, we used a backward selection procedure, thus removing effects if removal resulted in a lower Akaike Information Criterion (AIC) [76]. After fitting the multi-level model and selecting the appropriate demographic terms, we used the residuals (and not the raw scores) to decide whether the scores were abnormal. We used the median absolute deviation from the median (MAD) [77] instead of the more common approach of standard deviations because a few outlying scores can increase the standard deviation considerably. We used -3.5 MAD as a cut-off criterion, meaning all residuals exceeding -3.5 MAD were removed.

As the goal of this paper is to create regression formulas with which normative comparisons can be made, it is, therefore, necessary that the dependent variables are normally distributed [78]. Cognitive screening instruments such as the MoCA are not normally distributed and are usually left-skewed, also due to the effects of demographic variables [79]. One solution is to partial out the effects of demographic variables by means of using the residual scores from a regression analysis. However, these residual scores may still not be normal. We, therefore, chose to partial out the effects of the demographic variables and, as a next step, transformed all scores as this is recommended to meet the assumption of normality [80]. Instead of transforming the raw scores by common transformations, such as the square root of the raw scores, we used the Box–Cox procedure [81,82] to find the power transformation to best approximate the normal distribution. The Box–Cox procedure uses an algorithm that looks at several possible power transformations of the raw data (e.g., 0.501, 0.502, 0.503, etc.) and evaluates the distribution of the residuals with each power transformation. The transformation that results in the best approximation of normality for the residuals is saved (using an acceptable range of −2.0–+2.0 for skewness and −0.7–+0.7 for kurtosis [83]). By applying the Box–Cox-selected power transformation to the raw data, the residuals (from the previously selected models) are as normally distributed as possible [78]. The power transformations may result in very small or large values on the MoCA, which may be difficult to interpret. Therefore, we standardized all these transformed scores to the familiar *z*-scale with a mean of 0 and a standard deviation of 1. In the final step, the multi-level regression models were again fitted to obtain the parameter estimates for the clean, normalized dataset. As before, the AIC selects the demographic terms. The final model was then saved.

The resulting regression analyses were then used to compute the expected score (ES) for the transformed and standardized MoCA Total Score and MoCA-MIS and back-transformed to the original scoring scale of the MoCA Total Score and MoCA-MIS. Subsequently, residual scores (RS) were calculated for each individual by subtracting each individual’s expected score from the observed score (RS = OS − ES). The frequency distribution for the residue scores for the MoCA Total Score and MoCA-MIS was then converted into a percentile distribution [84].

All analyses were performed in RStudio [85].

## 3. Results

Figure 1 shows the flowchart of the data processing and the model fit, resulting in final samples of *n* = 820 for the MoCA total score and *n* = 182 for the MoCA-MIS, respectively. The Box–Cox power transformation with an exponent of 2.85 for the MoCA Total Score resulted in a skewness estimate of −0.091 and a kurtosis of 2.355. For the MoCA-MIS, Box–Cox power transformation with an exponent of 3.09 resulted in a skewness estimate of 0.157 and a kurtosis of 1.996, which are well within the acceptable range for normality.

For the MoCA Total Score, we used AIC model selection. The best-fit model included all transformed and standardized predictor variables, age, sex, and education, resulting in the following regression formula: −1.203 + 0.077 × sex − 0.01 × age + 0.208 × education (Conditional *R*^2^ = 0.4, Marginal *R*^2^ = 0.12, restricted maximum likelihood (REML) criterion at convergence: 3470.1). Table A1 in Appendix A shows the percentile distribution for the MoCA Total Score, stratified for age, education level, and sex.

For the MoCA-MIS, the same procedure was followed, and, in this case, also all predictor variables were included in the model, resulting in the following regression formula: −1.984 + 0.3 × sex − 0.011 × age + 0.270 × education (Marginal *R*^2^ = 0.14. REML criterion at convergence: 682.8). Random effect variances were not high enough to calculate *R*^2^ for random effects for the MoCA-MIS. Table A2 in Appendix A shows the regression-based percentile distributions for the MoCA MIS stratified for age group, education level, and sex.

## 4. Discussion

Here, we present regression-based normative data for the Dutch version of the MoCA and the MoCA-MIS based on a cognitively unimpaired sample of 820 individuals aged between 18 and 91. For both the MoCA Total Score and the MoCA-MIS, we found that age, educational attainment (using three categories: low, average, and high), and sex were significant predictors. The age-, education-, and sex-adjusted percentile distributions we present can be used for a fine-grained interpretation of MoCA(-MIS) scores, for instance, using recently published consensus criteria [86] (see Table 3). Such an approach is more valid than using the fixed cut-off score [4] that does not take age into account and which has only a limited adjustment for education level (i.e., one ‘bonus point’ awarded to individuals with 12 years of education or less). Furthermore, our wide age range also enables the administration and interpretation of MoCA scores in younger adults, which is relevant since the MoCA is increasingly being used in study samples or patients of a younger age and in disorders outside the field of mild cognitive impairment or dementia.

The finding that younger age and higher educational attainment were associated with a higher MoCA Total Score is in line with other international studies using the MoCA, most of which demonstrate considerable age and education effects (see Table 1). These results agree with evidence from most other neuropsychological tests, as the performance on those tests is consistently predicted by education level or years of education [87]. In addition, there is abundant evidence that cognitive function declines over time across the life span, especially in cognitive domains that represent fluent abilities such as executive function, working memory, and episodic memory [88], which are well represented in the MoCA. To date, sex differences on the MoCA Total Score have been reported, especially in larger samples (see also Table 1), in line with our results. In our sample, sex was also found to be a predictor for the MoCA-MIS, with women outperforming men. This corroborates large-scale studies showing sex differences in the memory domain, with women overall performing better on verbal memory tests, while men tend to perform better on spatial memory tasks [89]. However, as in our study sample, it should be noted that sex differences in cognitive functions are typically very small yet significant in large samples, while the variability in performance within each sex is overall substantially larger than the differences between the two sexes [90].

The results of our study also stress the problematic nature of applying a single cut-off score to determine whether an individual is cognitively impaired, thus ignoring the substantial effect of demographic variables on the test performance. This was also recently noted in a study by Engedal et al. [36]. In their study sample of 4780 cognitively unimpaired individuals over 70, a normal scoring range of 22–27 was found, with approximately half of their participants obtaining a score below the established cut-off score of 26 that is said to be indicative of cognitive impairment [4]. This is not only due to the design of the Engedal et al. [36] study, which consisted of older adults over the age of 70, as in our sample, we also found a substantial proportion of participants performing below the cut-off scores for cognitive impairment and dementia, even in the younger ages. Moreover, it is important to stress, as also outlined by Engedal et al. [36], that the original study by Nasreddine et al. [4] adopted a design fully different from ours, as their aim was not to publish demographically-adjusted normative data but to compare a group of healthy volunteers with individuals with MCI or Alzheimer’s dementia. It should also be noted that their sample of cognitively unimpaired was relatively small (*n* = 90, mean age 72.8, mean years of education 13.3) and that both the MCI and the dementia sample in that study had fewer years of education (12.3 and 10.0 respectively), possibly resulting in a too liberal cut-off score, resulting in poor specificity. In our study and those of others (also see [36] for a brief overview), a low education level may result in low MoCA scores in cognitively unimpaired individuals. This point is further illustrated by a recently published study [91] in a large and highly educated US sample (*n* = 3650), in which 26.8% of cognitively normal White participants and 57.9% of cognitively normal Black individuals were misclassified as being ‘cognitively impaired’ when applying the cut-off of 26. Clearly, the specificity of a single cut-off score for the MoCA is too low for validly substantiating clinical classifications, making it crucial to apply demographically-adjusted normative analyses when interpreting MoCA scores to minimize the risk of false-positive outcomes [56,91,92].

Our study is the first to present age-, education-, and sex-adjusted normative data for the MoCA-MIS in younger adults (i.e., <50 years of age). This score was introduced recently [61], demonstrating that the MoCA-MIS was promising in predicting the conversion from mild cognitive impairment to Alzheimer’s dementia. Others [63] demonstrated that the MoCA-MIS was able to distinguish between patients with Korsakoff’s amnesia, patients with other alcohol-related cognitive disorders, and individuals with alcohol use disorder without cognitive deficits. More research on the applicability of the MoCA-MIS is needed in other clinical groups. Furthermore, the psychometric properties of the MoCA-MIS, notably the test–retest reliability, are suboptimal, probably because of the skewed nature of this variable due to ceiling effects in cognitively unimpaired individuals [6]. Our analyses took the non-normality and skewness of this variable into account, making the presented normative data of the MoCA-MIS applicable for use in clinical samples. It should nevertheless be emphasized that the MoCA-MIS, which relies on the encoding, free and cued recall, and recognition of five words, is not a substitute for a more extensive memory assessment, which typically involves memory for word lists, pictures, spatial locations, or paired associates [79].

Our study also has some limitations. First, although our total sample size is large, our sample size is modest for the MoCA-MIS, as this optional part was administered only to a subset of participants. Furthermore, even larger data sets of cognitively unimpaired individuals for the MoCA exist. However, our regression-based statistical approach overcomes the problem of stratified norms that numbers tend to be small in some age and education strata, as the regression-based norm has been shown to require 2.5–5.5 times smaller samples compared to traditional norming [64]. Furthermore, most larger-scale data sets have a smaller age range, usually limited to older adults, rather than the full adult lifespan of our current sample. Additionally, all data were collected using the Dutch MoCA 7.1. Recently, MoCA 8.1 was introduced in the Netherlands, but the versions only differ slightly. That is, 7.1 uses the word *madelief* (daisy) in the memory subtest, while in 8.1, the word *lelie* (lily) is used, and in 7.1, the MIS cued recall and recognition tests are optional, while these are part of the standard administration procedure in 8.1. These minor alterations make our normative data set also applicable for the MoCA 8.1. Obviously, our data have been collected using the Dutch version of the MoCA in Dutch-speaking individuals from the Netherlands. Thus, caution is needed when applying normative data (or cut-off scores) obtained with one language version of the MoCA to other language versions altogether (which also pertains to applying the original Canadian French/English cut-off score [4] in other countries). Although the official available international versions of the MoCA were created with the goal of being equivalent, slight differences between MoCA performances across regions may be present. These may not only be due to variations between the respective translations but also related to the use of culture-specific items (for instance, different international MoCA versions use different animals for naming, related to the familiarity with these animals in a specific region of the world, also see [93]). Finally, although we applied exclusion criteria to exclude individuals with psychiatric disorders or brain diseases, we did not perform extensive neuropsychological assessments or obtain magnetic resonance images in our participants. As a result, there is the possibility that our sample of cognitively unimpaired people contains individuals with undetected or subclinical cognitive dysfunction or brain pathology. However, it should be stressed that only recruiting individuals without a history of any disease, pathology, symptom, or complaint will result in supernormal samples that are not representative of the general population [36].

## 5. Conclusions

In conclusion, our study results highlight the need for age-, education-, and sex-adjusted normative data for the MoCA. In addition to its Total Score, we also present normative data for the MIS, which is relevant for the screening of individuals with possible memory impairment. As the normative data presented here cover the full adult life span (from ages 18 to 91), they can be used for the screening of overall cognitive deficits and memory impairment, not only in older adults with (suspected) neurodegenerative disease but also in younger individuals with possible cognitive impairment.

## Figures and Tables

**Figure 1 jcm-11-04059-f001:**
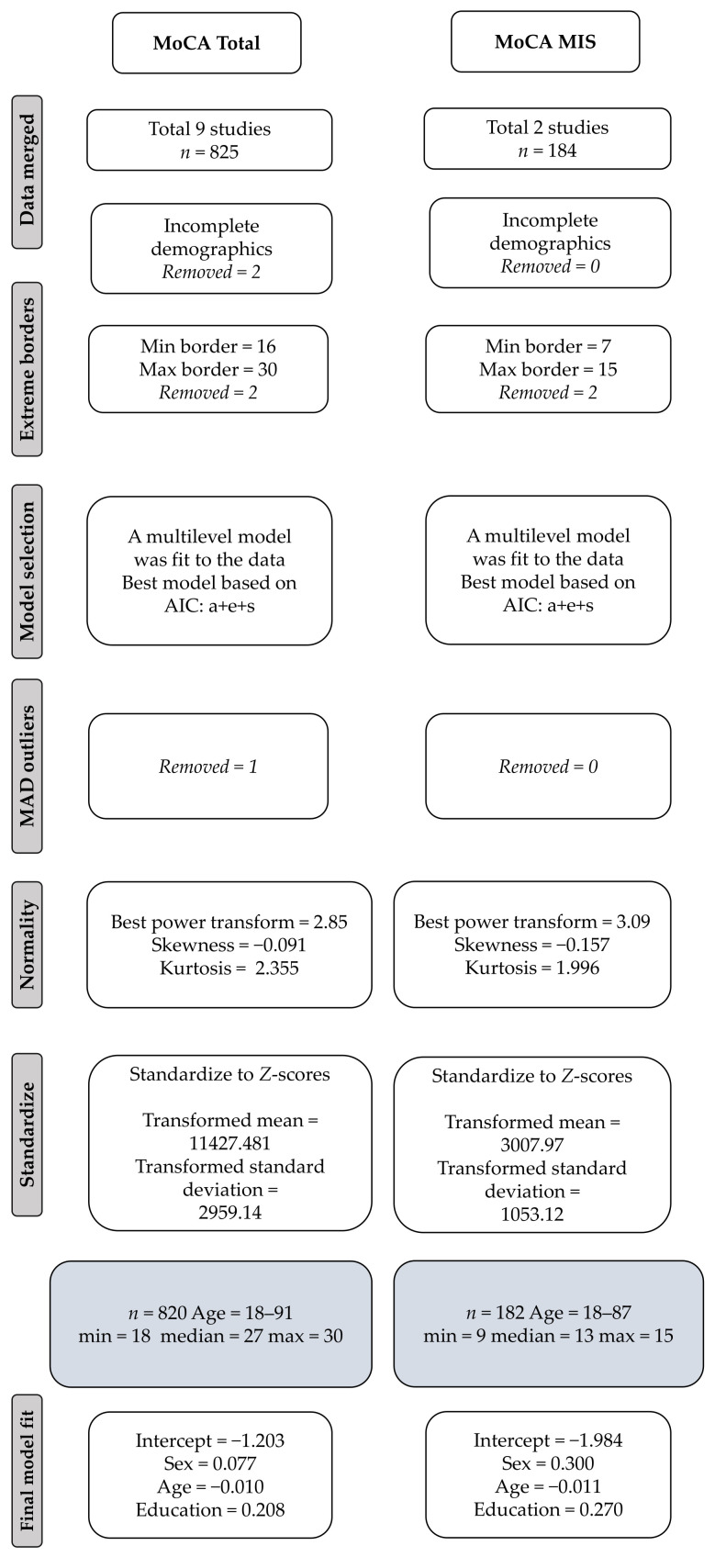
Flowchart showing the included studies and their data processing.

**Table 1 jcm-11-04059-t001:** Overview of available normative data for international versions of the Montreal Cognitive Assessment (MoCA).

Study	Country	*n*	Age Range	Adjustment	Method	MIS
Aiello et al. [29]	Italy	579	21–96	Age, Edu	Regression	No
Apolinario et al. [30]	Brazil	597	50–90	Age, Edu	Regression	Yes
Bartos and Fayette [31]	Czech Republic	1552	65–76	Age, Edu	Regression	No
Borland et al. [32]	Sweden	758	65–85	Age, Edu, Sex	Regression	No
Cesar et al. [33]	Brazil	385	60–80+	Age, Edu	Stratified	No
Classon et al. [34]	Sweden	181	80–94	Age, Edu	Regression	No
Conti et al. [35]	Italy	225	60–80	Age, Edu	Regression	No
Engedal et al. [36]	Norway	4780	70–90	Age, Edu, Sex	Regression	No
Freitas et al. [37]	Portugal	650	25–91	Age, Edu	Regression	No
Gaete et al. [38]	Chile	526	18–90	Age, Edu	Regression	No
Gluhm et al. [39]	USA	254	20–89	Age	Stratified	No
Hayek et al. [40]	Lebanon	164	60–87	Age, Edu	Regression	No
Ihle-Hansen et al. [41]	Norway	3413	63–65	Edu	Mean + SD	No
Kang et al. [42]	Korea	211	60–90	Age, Edu, Sex	Stratified	No
Kenny et al. [43]	Ireland	5802	50–85	Age, Edu	GAMLSS	No
Konstantopoulos et al. [11]	Greece	710	20–85	Age, Edu, Sex	Regression	No
Kopecek et al. [44]	Czech Republic	540	60–98	Age, Edu	Regression	No
Larouche et al. [45]	Canada	1019	41–98	Age, Edu, Sex	Regression	No
Lee et al. [46]	Korea	115	69.1 (±6.1) ^†^	Edu ^‡^	Mean + SD	No
Lu et al. [47]	China	6283	65–100	Age, Edu	Stratified	No
Malek-Ahmadi et al. [48]	USA	205	70–99	Age, Edu	Stratified	No
Muayqil et al. [49]	Saudi Arabia	311	18–80	Age, Edu	Stratified	No
Narazaki et al. [50]	Japan	1977	65–96	Age, Edu	Regression	No
Nasreddine et al. [4]	Canada	90	72.8 (±7.0) ^†^	Edu ^‡^	Mean + SD	No
Ojeda et al. [51]	Spain	700	18–86	Age, Edu	Regression	No
Pereiro et al. [52]	Spain	563	50–97	Age, Edu	Regression	No
Pinto et al. [53]	Brazil	110	65–88	Age, Edu	Regression	No
Rossetti et al. [54]	USA	2653	18–85	Age, Edu	Stratified	No
Rossetti et al. [55]	USA	1118	18–75	Age, Edu	Stratified	No
Sachs et al. [56]	USA	5338	55–85	Age, Edu, Sex, Race	Regression	No
Santangelo et al. [57]	Italy	415	21–95	Age, Edu	Regression	No
Serrano et al. [8]	Argentina	155	60–91	Age, Edu	Stratified	No
Siciliano et al. [58]	Italy	302/413 ^#^	20–89	Age, Edu	Regression	No
Sink et al. [59]	USA	414	35–83	Age, Edu	Stratified	No
Thomann et al. [60]	Switzerland	283	65–91	Age, Edu, Sex	Regression	No
*Current Study*	*Netherlands*	*820*	*18–91*	*Age, Edu, Sex*	*Regression*	*Yes*

MIS = Memory Index Score reported; ^#^ MoCA parallel versions 2 and 3 respectively; ^†^ only mean age + SD reported; ^‡^ 1 point added for 12 years of education or less; GAMLSS = Generalized Additive Models for Location Scale and Shape.

**Table 2 jcm-11-04059-t002:** Number of individuals per age group, stratified for education (into three groups) and sex (for descriptive purposes).

Education	Low		Average		High	
Age Group	M	W	M	W	M	W
18–29	0	0	33	54	30	65
30–39	4	0	11	22	22	20
40–49	7	2	19	29	15	27
50–59	21	9	22	27	39	25
60–69	16	28	25	29	43	31
70–79	9	20	13	23	25	27
≥80	4	7	2	8	5	6

Education level is based on the Dutch educational system (7 categories) [66], years of Education approximate equivalent according to the Anglo-Saxon educational system [67]. Low = levels 1–4 (≤9 years of education), Average = level 5 (10–11 years of education), high = levels 6–7 (≥12 years of education). M = men, W = women.

**Table 3 jcm-11-04059-t003:** Consensus criteria for converting percentiles to diagnostically meaningful labels (based on [85]).

Percentile	Diagnostic Label
≥98	Exceptionally high
91–97	Above average
75–90	High average
25–74	Average
9–24	Low average
2–8	Below average
<2	Exceptionally low

## Data Availability

The normative data reported in this study are part of the Advanced Neuropsychological Diagnostics Infrastructure (ANDI), available at www.andi.nl (accessed on 20 June 2022).

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
