# Peer review of "Regression-Based Normative Data for the Montreal Cognitive Assessment (MoCA) and Its Memory Index Score (MoCA-MIS) for Individuals Aged 18–91"

_jcm, 2022, doi:10.3390/jcm11144059_

Round 1

Reviewer 1 Report

This large cohort study computed the normative data for the Dutch-language version of the MoCA and the MoCA-MIS for individuals aged 18-91. The main findings of this study were the need for age, education, and sex adjusted normative data. I feel that this study had a large impact on the clinicians and had been written appropriately. However, this manuscript needs some substantial modification for a better understanding. My major concerns are as follows:

1.     Abstract: Please add (2).

2.     Appendix: What do you show in the appendixes? Please add captions to those.
If the results were true, some healthy young adults with high education may have MCI (MoCA<26). Is it true?

3.     Appendix: Please add the number of subjects classified by each education level in each age group. I think a few subjects were included in younger adults (age: 18-29) with low education level.

4.     Abstract and Results: I believe it is better if the typical normative data is written or summarized in the results section and the abstract.

Author Response

This large cohort study computed the normative data for the Dutch-language version of the MoCA and the MoCA-MIS for individuals aged 18-91. The main findings of this study were the need for age, education, and sex adjusted normative data. I feel that this study had a large impact on the clinicians and had been written appropriately.

We thank the reviewer for these positive words, highlighting the potential impact of our manuscript for clinicians.

However, this manuscript needs some substantial modification for a better understanding. My major concerns are as follows:

  1. Abstract: Please add (2).

We have corrected this typo.

  1. Appendix: What do you show in the appendixes? Please add captions to those.

We have added a general caption for Appendix A. We also provide a scoring example for an individual case, see lines 411-419: “The tables in this appendix show the percentile distribution that allows for age, education and sex adjustment. Although the presentation of the tables is stratified according to age, education level and sex, this was done for presentation purposes only, as the percentile distribution is based on the outcome of the multilevel regression analysis (taking into account the sometimes uneven distribution of individuals across the individual strata). These tables can be used to convert an individual’s MoCA Total Score or Memory Index Score (MIS) into an age-, education- and sex-adjusted percentile equivalent. The individual’s percentile can be used for clinical interpretation, for example using the nomenclature described in Table 3 [86].
For instance, a female patient aged 64 with a university degree obtains a MoCA Total Score of 22 and a MIS of 8. This results in percentile equivalent of 4 for the MoCA Total Score and a percentile of 1 for the MIS, reflecting a ‘below average’ overall cognitive performance and an ‘exceptionally low’ memory performance.”

If the results were true, some healthy young adults with high education may have MCI (MoCA<26). Is it true?

Note that this is exactly why fine-grained normative data are needed for the MoCA, and also in line with results of other normative data sets for the MoCA. That is, the widely-used cut-off score of 26 is based on the study by Nasreddine et al. (2005), which included only 90 older controls (mean age=72.8, mean years of education=13.3), whose MoCA performance was contrasted with the performance of a mild cognitive impairment (n=94) and an Alzheimer’s dementia (n=93) group – both with less years of education than the normal controls – using receiver operating characteristics (ROC) analyses. As outlined by Engedal at al. (2022), cited as [36], the Nasreddine design differs from the approach they (and also we in our present paper) used, in that “The aim was not to examine normative scores for the healthy volunteers but to examine which score on the MoCA best distinguished the three groups. Thus, the setting, design, and purpose of the study by Nasreddine et al. differs significantly from the design of the present study and other population-based normative studies on the MoCA” (pp. 595-596). Thus, the use of the cut-off score of 26 results in a too high number of false positive outcomes (i.e. individuals with an ‘impaired’ screening score but without actual cognitive impairment), illustrated by the fact that in the Engedal study, half of their participants performed below this cut-off score. This point is also substantiated by a study that has just been published using the MoCA in a large US sample (Ratcliffe et al., The Clinical Neuropsychologist, https://doi.org/10.1080/13854046.2022.2086487, cited as 91), which showed that using the cut-off of 26, 26.8% of White cognitively normal and no less than 57.9% of Black cognitively normal individuals were misclassified as being ‘cognitively impaired’. These findings stress the need for demographically-adjusted normative data (as also outlined in the conclusion by Ratcliffe and colleagues.
We have expanded the Discussion section, addressing this issue and also citing the new Ratcliffe et al. paper [91] and adding a reference to a study also highlighting the poor specificity of the MoCA cut-off score (Thomann et al., reference [92]
), see lines 301-318: “Moreover, it is important to stress, as also outlined by Engedal et al. [36], that the original study by Nasreddine et al. [4] adopted a design fully different from ours, as their aim was not to publish demographically-adjusted normative data, but to compare a group of healthy volunteers with individuals with MCI or Alzheimer’s dementia. It should also be noted that their sample of cognitively unimpaired was relatively small (n=90, mean age 72.8, mean years of education 13.3), and that both the MCI and the dementia sample in that study had less years of education (12.3 and 10.0 respectively), possibly resulting in a too liberal cut-off score, resulting in poor specificity. In our study and those of others (also see [36] for a brief overview), a low education level may result in low MoCA scores in cognitively unimpaired individuals. This point is further illustrated by a recently published study [91] in a large and highly educated US sample (n=3,650), in which 26.8% of cognitively normal White participants and 57.9% of cognitively normal Black individuals were misclassified as being ‘cognitively impaired’ when applying the cut-off of 26. Clearly, the specificity of a single cut-off score for the MoCA is too low for validly substantiating clinical classifications, making it crucial to apply demographically-adjusted normative analyses when interpreting MoCA scores to minimize the risk of false-positive outcomes [56,91,92].”

  1. Appendix: Please add the number of subjects classified by each education level in each age group. I think a few subjects were included in younger adults (age: 18-29) with low education level.

We have revised Table 2 on p. 4, now indicating the number of individuals per age group, stratified for education level and sex. Note that the regression-based approach takes into account lower numbers of individuals in some strata (as opposed to a stratified normative approach), for which we have also added a reference [64]. The tables presented in Appendix A thus reflect the outcome of the regression analyses; the stratified presentation of the percentile distribution is only done out of convenience to enable an easy conversion to percentile scores using these published tables, as using the continuous data for age, education and sex would require the use of software to compute percentiles (note that this is why the online ANDI infrastructure was established, in which these norms are incorporated), also discussed in reference [65].  This is also outlined in the Discussion, lines 338-341: “However, our regression-based statistical approach overcomes the problem of stratified norms that numbers tend to be small in some age and education strata, as regression-based norm has been shown to require 2.5-5.5 times smaller samples compared to traditional norming [64].” We also expanded the introduction with a reference to the Oosterhuis et al. [64] and Heaton et al. [65] paper on demographically-adjusted regression-based normative data: “The present study sets out to compute age- and education based normative data for the Dutch-language version of the MoCA and the MoCA-MIS for individuals aged 18-91 using a demographically-adjusted regression-based normative approach, the validity of which has long been established for neuropsychological tests [64, 65]. Such normative data facilitate the use of the MoCA as a short, but valid cognitive screen in patient samples other than older adults with MCI or dementia.“ (lines 125-130).

  1. Abstract and Results: I believe it is better if the typical normative data is written or summarized in the results section and the abstract.

We have added the transformation details and regression formulas to the Results section (lines 241-256): “The Box-Cox power transformation with an exponent of 2.85 for the MoCA Total Score resulted in a skewness estimate of -0.091 and a kurtosis of 2.355. For the MoCA-MIS, Box-Cox power transformation with an exponent of 3.09 resulted in a skewness estimate of 0.157 and a kurtosis of 1.996, which are well within the acceptable range for normality. For the MoCA Total Score we used AIC model selection. The best-fit model included all transformed and standardized predictor variables; age, sex, and education resulting in the following regression formula: -1.203 + 0.077×sex – 0.01×age + 0.208×education (Conditional R2 =0.4, Marginal R2 = 0.12, restricted maximum likelihood (REML) criterion at convergence: 3470.1). Table 1A in Appendix A shows the percentile distribution for the MoCA Total Score, stratified for age, education level and sex. For the MoCA-MIS, the same procedure was followed and, in this case, also all predictor variables were included in the model, resulting in the following regression formula: -1.984 + 0.3×sex – 0.011×age + 0.270×education (Marginal R2 =0.14. REML criterion at convergence: 682.8). Random effect variances were not high enough to calculate R2 for random effects for the MoCA-MIS. Table 2A in Appendix A shows the regression-based percentile distributions for the MoCA MIS stratified for age group, education level and sex.” We have also added the model-fit results to the abstract (lines 28-30). The normative data themselves are presented in the Appendix A, which cannot be summarized in the abstract. They are presented as an Appendix rather than in the Results section, as these tables are large and would complicate the formatting of the body text of the manuscript.

Reviewer 2 Report

1.The introduction is very short.

2.The Procedure and statistical analyses are not accurately described. The authors should better explain this section.

3.The discussion is short, as is the introduction. By expanding the first, the second will necessarily be expanded.

Author Response

1.The introduction is very short.

The Introduction section of our 18-page manuscript has a length of 1078 words, and in addition also includes a comprehensive table summarizing previous findings from normative studies on the MoCA (Table 1) and the Discussion has a length of 1437 words, not including the Conclusion. We are a bit puzzled by the reviewer’s comment, as these are normal lengths for a paper written for a biomedical journal such as JCM. We have checked the 8 most recently published empirical papers in JCM[1], and found that our manuscript is even on the lengthy side (we found introductions with a range of 284-927 words, with a median of 417 words, and introductions with a range of 559-2044 words, with a median of 1138 words). As our introduction and discussion adhere to the guidelines for authors to be “concise and comprehensive”, we feel that adding words for the mere goal of creating wordiness will not improve the readability of our manuscript. Note that we have added more information on the rationale behind regression-based norms to the revised discussion (lines 125-130).

2.The Procedure and statistical analyses are not accurately described. The authors should better explain this section.

We have expanded the Materials and Methods section with more details on the educational coding, creation of the normative tables, transformation, normality checks and standardization, procedure, and statistical package used. See lines 154-158: “Education level was scored in 7 categories, based on the Dutch educational system [66] that is similar to UNESCO’s ISCED scale [73], in which 1 reflects less than primary school and 7 an academic degree. These levels were then categorized as low (levels 1-4), average (level 5) or high (levels 6-7) for descriptive purposes only.”; lines 162-163: “All participants were administered the authorized paper-and-pencil Dutch version of the Montreal Cognitive Assessment 7.1”; lines 182-183: “Age was coded as male=0 and female=1, education was coded in the aforementioned 7 categories.”; lines 219-225: “The transformation that results in the best approximation of normality for the residuals is saved (using an acceptable range of -2.0 − +2.0 for skewness and -0.7 − +0.7 for kurtosis [83]). By applying the Box-Cox-selected power transformation to the raw data, the residuals (from the previously selected models) are as normally distributed as possible [78]. The power transformations may result in very small or large values on the MoCA, which may be difficult to interpret. Therefore, we standardized all these transformed scores to the familiar z-scale with mean of 0 and standard deviation of 1.”; lines 229-237 “The resulting regression analyses were then used to compute the expected score (ES) for the transformed and standardized MoCA Total Score and MoCA-MIS and back-transformed to the original scoring scale of the MoCA Total Score and MoCA-MIS. Subsequently, residual scores (RS) were calculated for each individual by subtracting each individual’s expected score from the observed score (RS = OS – ES). The frequency distribution for the residue scores for the MoCA Total Score and MoCA-MIS were then converted into a percentile distribution [84]. All analyses were performed in RStudio [85].” We also revised Figure 1, adding the standardization step that was not presented in the original flow-chart (p. 6). Note that the statistical procedures of the ANDI infrastructure databased have already been described previously in de Vent et al. [74], and that a more comprehensive – and more statistically detailed – description is beyond the scope of this manuscript. We refer the reader to this paper in lines 173-174: “More details about this process can be found elsewhere [74], but we will provide a brief summary of all the data processing steps here.”

3.The discussion is short, as is the introduction. By expanding the first, the second will necessarily be expanded.

We have expanded the Discussion section, also citing new studies highlighting the poor specificity of the MoCA cut-off score, see lines 301-318: “Moreover, it is important to stress, as also outlined by Engedal et al. [36], that the original study by Nasreddine et al. [4] adopted a design fully different from ours, as their aim was not to publish demographically-adjusted normative data, but to compare a group of healthy volunteers with individuals with MCI or Alzheimer’s dementia. It should also be noted that their sample of cognitively unimpaired was relatively small (n=90, mean age 72.8, mean years of education 13.3), and that both the MCI and the dementia sample in that study had less years of education (12.3 and 10.0 respectively), possibly resulting in a too liberal cut-off score, resulting in poor specificity. In our study and those of others (also see [36] for a brief overview), a low education level may result in low MoCA scores in cognitively unimpaired individuals. This point is further illustrated by a recently published study [91] in a large and highly educated US sample (n=3,650), in which 26.8% of cognitively normal White participants and 57.9% of cognitively normal Black individuals were misclassified as being ‘cognitively impaired’ when applying the cut-off of 26. Clearly, the specificity of a single cut-off score for the MoCA is too low for validly substantiating clinical classifications, making it crucial to apply demographically-adjusted normative analyses when interpreting MoCA scores to minimize the risk of false-positive outcomes [56,91,92].”  With respect to the length of the Discussion, we kindly refer to our reply to the first comment of this reviewer.

Notes:

[1] https://doi.org/10.3390/jcm11133878; https://doi.org/10.3390/jcm11133877; https://doi.org/10.3390/jcm11133875; https://doi.org/10.3390/jcm11133874; https://doi.org/10.3390/jcm11133873; https://doi.org/10.3390/jcm11133872; https://doi.org/10.3390/jcm11133870; https://doi.org/10.3390/jcm11133869

Round 2

Reviewer 2 Report

Dear Authors, 

thank you for the changes made to the manuscript.